# Antioxidant and Anti-Inflammatory Effects of 6,3’,4´- and 7,3´,4´-Trihydroxyflavone on 2D and 3D RAW264.7 Models

**DOI:** 10.3390/antiox12010204

**Published:** 2023-01-16

**Authors:** Xiang Wang, Yujia Cao, Siyu Chen, Xin Yang, Jinsong Bian, Dejian Huang

**Affiliations:** 1Department of Food Science and Technology, National University of Singapore, 2 Science Drive 2, Singapore 117542, Singapore; 2National University of Singapore (Suzhou) Research Institute, 377 Linquan Street, Suzhou 215123, China; 3Department of Pharmacology, School of Medicine, Southern University of Science and Technology, Shenzhen 518055, China

**Keywords:** 6,3´,4´-trihydroxyflavone, 7,3´,4´-trihydroxyflavone, cellular-Src tyrosine kinase, antioxidant, anti-inflammation, 3D macrophage model

## Abstract

Dietary flavones 6,3´,4´-trihydroxyflavone (6,3´,4´-HOFL) and 7,3´,4´-trihydroxyflavone (7,3´,4´-HOFL) showed preliminary antioxidant and anti-inflammatory activities in a two-dimensional (2D) cell culture model. However, their action mechanisms remain unclear, and the anti-inflammatory activities have not been studied in a reliable three-dimensional (3D) cell model. Therefore, in the current study, the antioxidant potency was examined by their scavenging ability of cellular reactive oxygen species. Anti-inflammatory activities were examined via their inhibitory effects on inflammatory mediators in vitro on 2D and 3D macrophage models, and their mechanisms were determined through transcriptome. In the 3D macrophages, two flavones were less bioactive than they were in 2D macrophages, but they both significantly suppressed the overexpression of proinflammatory mediators in two cell models. The divergent position of the hydroxyl group on the A ring resulted in activity differences. Compared to 6,3´,4´-HOFL, 7,3´,4´-HOFL showed lower activity on NO, IL-1β suppression, and c-Src binding (IC_50_: 12.0 and 20.9 µM) but higher ROS-scavenging capacity (IC_50_: 3.20 and 2.71 µM) and less cytotoxicity. In addition to the IL-17 and TNF pathways of 6,3´,4´-HOFL, 7,3´,4´-HOFL also exerted anti-inflammatory activity through JAK-STAT, as indicated by the RNA-sequencing results. Two flavones showed prominent antioxidant and anti-inflammatory activities on 2D and 3D models.

## 1. Introduction

Oxidative stress plays an essential role in cellular redox-dependent signaling transductions to regulate cell behaviors and feedback to the host to clear reactive oxygen species (ROS) in regulated cellular conditions. However, a sustained high level of ROS leads to inflammation via releasing inflammatory mediators and activating macrophages [1]. Inflammation is an immune response to stimulants to protect hosts from pathogens; however, dysregulated inflammation is deleterious and leads to chronic inflammation-related diseases such as cardiovascular and neurodegenerative diseases and even cancer. These diseases have been recognized as the main cause of death worldwide and a constituent threat to global human health and welfare [2]. To scavenge ROS and alleviate inflammation, the application of dietary bioactive compounds, especially flavones, has attracted widespread attention [3].

Among the dietary flavonoids, 6,3´,4´-HOFL (6,3´,4´-trihydroxyflavone) and 7,3´,4´-HOFL (7,3´,4´-trihydroxyflavone) are the simplest but relatively rare natural flavones aglycones. 7,3´,4´-HOFL was first extracted from *Trifolium repens* and *Medicago sativa* in the 1960s, but 6,3´,4´-HOFL was extracted from *Dipteryx lacunifera* in 2020 [4]. These legumes were consumed as food or valuable herbs in folk medicines across the world. For example, *T. repens* has been used to treat dermal, stomach, lung, and nervous disorders in European and American cultures [5]. *D. lacunifera* is a native plant in the northeast of Brazil, and its seeds were consumed as snacks by local residents [4]. Both flavones have been reported to be bioactive compounds with anticancer and antioxidant activities [4]. The anti-inflammatory activity of 7,3´,4´-HOFL was suggested by nitric oxide (NO)-inhibiting activity, [6], but its signaling pathway has not been clearly revealed yet.

The traditional anti-inflammatory agents’ screening model is cells cultured on polyester or the glass bottom of plates. Although it is a cornerstone in cell biological research, the 2D surface-forced cells flatten on the bottom in a monolayer, leading to abnormal cell morphology and impeding cell–cell and cell–extracellular matrix (ECM) communications. The divergent microenvironments from real tissue result in different cellular behaviors, such as proliferation rate [7], gene expressions [8], and drug resistance [9]. The intrinsic defectives of the 2D model have led to a high failure rate in in vitro-to-in vivo translation in drug development [10]. Furthermore, animal or clinical studies have high costs and are complex and involved in ethical controversy, so a bridge between in vitro and in vivo studies is in urgent demand. Three-dimensional (3D) cell models mimic in vivo microenvironments to narrow the gap and speed up the discovery of bioactive compounds for human health promotion [11]. Electrohydrodynamic (EHDJ)-printed poly(ε-caprolactone) (PCL) scaffolds are biocompatible, lowly immunogenic, optimally biodegradable, and have comparable pore and fiber size (approximately 10 µm) matching macrophage scale [12]. The 3D macrophages show higher lipopolysaccharide (LPS) sensitivity and lower anti-inflammatory agent sensitivity, delivering higher physiological relevance [13].

Herein, we report the discovery of a comparative study of the antioxidant and anti-inflammatory activity of 6,3´,4´-HOFL and 7,3´,4´-HOFL and explore the action mechanisms on 2D and 3D macrophages.

## 2. Materials and Methods

### 2.1. Materials

6,3´,4´-HOFL and 7,3´,4´-HOFL were purchased from INDOFINE Chemical Company, Inc. (Hillsborough Township, NJ, USA). Bio-grade DMSO (purity ≥ 99.9%), LPS (from Escherichia coli serotype 055:B5, purity ≥ 99%), c-Src Inhibitor Screening Kit, 2´,7´-dichlorofluorescin diacetate (DCFDA) (purity ≥ 97%), bisbenzimide H33342 (purity ≥ 97%), and tert-butyl hydroperoxide (tBHP) were purchased from Sigma-Aldrich Co., Ltd. (Singapore). High-glucose Dulbecco’s Modified Eagle Medium (DMEM), penicillin–streptomycin antibiotics (1000 IU/ mL penicillin and 1000 µg/mL streptomycin), fetal bovine serum (FBS), phosphate-buffered saline (PBS), BCA Protein Assay Kit, radioimmunoprecipitation (RIPA) lysis buffer, and TRIzol reagent were supplied by Thermo Fisher Scientific Co., Ltd. (Singapore). Cell counting kit-8 (CCK-8) was purchased from Dojindo Molecular Technologies, Inc. (Kumamoto, Japan). Griess reagent and Gotaq^®^ qPCR Master Mix were purchased from Promega Pte Ltd. (Madison, Wisconsin, DC, USA). F-actin staining kit red fluorescence cytopainter was purchased from Abcam (Singapore). Ultralow attachment multi-well plates were supplied by Corning Inc. (New York, NY, USA). All the antibodies were purchased from Cell Signaling Technology Inc. (Danvers, MA, USA). The iScript cDNA synthesis kit and polyvinylidene fluoride (PVDF) membranes were supplied by Bio-Rad Laboratories Pte. Ltd. (Singapore). All the primers were purchased from Integrated DNA Technologies Pte. Ltd. (Singapore).

### 2.2. Scaffolds Fabrication

A home-built EHDJ printer was used to fabricate PCL scaffolds with 100 μm pore size, as described in our previous study (Appendix A) [12]. Prior to seeding cells, scaffolds were sterilized by sequential treatment of ultraviolet light, soaking in 70% ethanol, 1000 ppm bleach for 15 min and finally rinsing by deionized water.

### 2.3. 2D and 3D Cell Culture and Morphology Characterization

The RAW264.7 cell lines (TIB-71, ATCC, Manassas, VA, USA) were maintained in high-glucose DMEM with 10% FBS and 1% penicillin–streptomycin antibiotics at 37 °C with 5% CO_2_. The 3D macrophage model was established by gently adding cell suspensions with a density of 1.0 × 10^6^ cells/mL on the scaffolds in ultralow attachment plates. DMEM was gradually added within 5 h to promote cell attachment on fibers.

The morphology of 2D and 3D cells was observed using FESEM (field emission scanning electron microscope) and CLSM (confocal laser scanning microscopy). Cells grown on plates and scaffolds were fixed with 4% paraformaldehyde. Samples were dehydrated by gradient ethanol solution (10−100% *v*/*v*), dried in a vacuum oven at 40 °C overnight, and, finally, coated with platinum before being observed by FESEM (JSM 183 6701F, JEOL, Tokyo, Japan). Samples for CLSM observation were stained with 10 μg/mL H33342 and F-actin staining kit red fluorescence cytopainter to visualize the cell nuclei and F-actin, respectively. The 2D and 3D cell images were captured under CLSM (LSM 710, ZEISS, Germany) in the dark. A 3D reconstructed z-stack confocal image was processed using ZEN software blue edition.

### 2.4. Cellular Antioxidant Analysis

The cellular antioxidant activities were tested by measuring intracellular ROS detected using H_2_DCFDA fluorescent probe. Macrophages were pretreated with 6,3´,4´- or 7,3´,4´-HOFL at various concentrations for 4 h, and then, they were stressed with tBHP for 20 h. Next, H_2_DCFDA (20 µM) was incubated with cells to visualize cellular ROS. The endpoint ROS was measured at Ex/Em 485/528 nm using Synergy HTX Multi-Mode Reader (BioTek). For the fluorescent images of cells, their nuclei were stained with 10 µg/mL H33342 and observed under CLSM in the dark.

### 2.5. Cell Viability Assay

Cell viability was assessed using cell counting kit-8 (CCK-8). After cells were incubated with 6,3´,4´-HOFL or 7,3´,4´-HOFL at different concentrations for 24 h, the CCK-8 working solution was added and absorbances were measured following manufacturer’s instructions. The results were converted to the percentage of untreated cells (control group), and each result was performed in three independent experiments in duplicate.

### 2.6. NO Production Assay

After 2D and 3D cells were pretreated with 6,3´,4´-HOFL or 7,3´,4´-HOFL at different concentrations for 4 h and then with 100 ng/mL LPS for 20 h, nitrite (NO_2_^-^) concentrations were measured following the manufacturer’s protocol. Absorbance at 540 nm was detected and results were normalized to the percentage of LPS-treated cells (LPS group).

### 2.7. Western Blotting Analysis

2D and 3D macrophages in 6-well plates were pretreated with 6,3´,4´- or 7,3´,4´-HOFL at a concentration of 22.1 and 26.7 μM for 4 h, before the stimulation with LPS (100 ng/mL) for 20 h. Total proteins were extracted and resolved, and Western blotting was conducted as commonly reported (Fu et al., 2022). Inducible nitrite oxidase (iNOS) and cyclooxygenase-2 (COX-2) bands were quantified using ImageJ, and data were expressed as fold compared to the 2D LPS group.

### 2.8. RT-qPCR

Total RNA was extracted from 2D and 3D cells using TRIzol reagent following the supplier’s protocol with modifications. Chloroform was applied to wash away PCL scaffolds dissolved in TRIzol (Wang et al., 2021b). The total RNA was reverse-transcribed using an iScript cDNA synthesis kit on Thermo Cycler (T100, Bio-Rad, Singapore). As templates, cDNA was mixed with gene primers and qPCR master mix, and then, the mixture was applied to a StepOnePlus Real-Time PCR System (Applied Biosystem, Singapore) to conduct real-time quantitative polymerase chain reaction (qPCR).

β-actin was used as the internal reference gene to determine other gene expression changes, and the results were expressed as 2^−ΔΔCt^. Primer sequences are attached in Appendix A.

### 2.9. Screening for C-Src Kinase Binding Activity

The dose–response curves of 6,3´,4´- and 7,3´,4´-HOFL in inhibiting human c-Src activity were measured using a c-Src Inhibitor Screening Kit. Stock solutions of the two flavones at the concentration of 500 µM were prepared in the assay buffer and diluted to a range of concentrations. Other assay reagents were added following the supplier’s protocol, and the amount of ADP was quantified by the fluorescence intensity monitored at Ex/Em 535/587 nm on the microplate reader (BioTek, Synergy HTX, USA) at 37 °C. The relative inhibition percentages were calculated as following equations:R_TF_ = [(RFU_TF2_ − RFU_TF1_) − (RFU_BC2_ − RFU_BC1_)]/(T_2_ − T_1_)(1)
R_NI_ = [(RFU_NI2_ − RFU_NI1_) − (RFU_BC2_ − RFU_BC1_)]/(T_2_ − T_1_)(2)
% Relative Inhibition = (R_NI_ − R_TF_)/ R_NI_ × 100%(3)

T_1_ = 1500 s, T_2_ = 1900 s

R: reaction rate; TF: test flavones; BC: background; NI: no inhibitor.

The crystal structure of human tyrosine-protein kinase c-Src was downloaded from the Protein Data Bank (PDB ID: 2SRC). AMP-PNP and all unnecessary water molecules were removed, and hydrogen atoms were added. The structures of 6,3´,4´- and 7,3´,4´-HOFL were drawn using ChemDraw Prime 17.1, and their energies were minimized on Chem3D Ultra 12.0. PDB files of c-Src and 6,3´,4´- or 7,3´,4´-HOFL were imported as receptors and ligands into AutoDock Vina to conduct docking. The results were visualized and processed on PyMol 2.3. The images were shown as the docked 6,3´,4´- and 7,3´,4´-HOFL in the ATP binding pocket showing hydrogen bonds with amino acid residues within 3 Å.

### 2.10. Transcriptome Sequencing

2D-cultured RAW264.7 cells were divided into Ctrl, LPS, HOFL_6, and HOFL_7. Ctrl was cells without any treatment, LPS was cells treated with 100 ng/mL LPS, HOFL_6 and HOFL_7 were pretreated with 6,3´,4´- (25 μM) or 7,3´,4´-HOFL (30 μM) for 4 h, respectively, and challenged with LPS. Library construction and the RNA-sequencing were performed by NovogeneAIT Genomics Singapore Pte. Ltd.

The libraries were constructed on a high-throughput Illumina platform and transformed to sequenced reads. Clean reads were obtained and aligned with HISAT2 (2.0.5) to Mus Musculus. The mapped reads were calculated to gene expression levels as FPKM (fragments per kilobase of transcript sequence per million base pairs sequenced). Differentially expressed genes (DEGs) were screened using the threshold of padj ≤ 0.05 and |log2 (fold change)| ≥ 1. The Kyoto encyclopedia of genes and genomes (KEGG) enrichment of DEGs was performed on clusterProfiler (3.8.1) to identify significantly (KEGG terms padj < 0.05) enriched signaling pathways of DEGs.

RT-qPCR was performed to confirm the RNA-sequencing results and to determine changes in the expression levels of key genes. Primer sequences are shown in Appendix A.

### 2.11. Statistical Analysis

Statistical analysis was performed using GraphPad Prism (version 8.0.2). Data comparisons were performed using one-way analysis of variance. Duncan’s correction was used for post hoc tests. *p* * < 0.05, *p* ** < 0.01, and *p* *** < 0.001 were used to indicate the statistical significance, and *p* * < 0.05 was considered statistically significant.

## 3. Results

### 3.1. Cellular Antioxidant Activity

The treatment of tBHP dramatically induced cellular ROS, and the pretreatment of 6,3´,4´- and 7,3´,4´-HOFL suppressed the ROS content (Figure 1). 6,3´,4´- and 7,3´,4´-HOFL significantly downregulated cellular ROS in a dose-dependent manner, and their IC_50_ values were 3.02 and 2.71 µM, respectively. The comparative IC_50_ values of ROS-scavenging ability suggest their antioxidant activity is not sensitive to the position of the hydroxyl group (Figure 1e). Furthermore, the cellular protective activity of two flavone isomers was verified by improved cell viability. The treatment with tBHP resulted in a 40% reduction in cell viability, but this was recovered by the treatment of 6,3´,4´- or 7,3´,4´-HOFL. As shown in Figure 1g, the cell viability recovered in a dose-dependent manner and reached over 80% at 20 µM of 6,3´,4´-HOFL and 50 µM of 7,3´,4´-HOFL. Notably, when the concentration of 6,3´,4´-HOFL was increased to 50 µM, the cell viability started to decrease, which may be attributed to its cytotoxicity at high dosages.

### 3.2. Morphologies of 2D and 3D Macrophages

On normal plates, macrophages attached to rigid surfaces showing flat shapes, and they were monolayer with one side facing the plate surface. This side of cells communicated with neither other cells nor with the medium and the components that dissolved in it, including nutrients, stimulants, and flavones. In contrast, macrophages on 3D scaffolds were completely different. They showed spherical shapes with multilayers resembling in vivo cells. Except for those attaching to fibers, cells in scaffolds were surrounded by other cells rather than rigid surfaces. This provided an opportunity for cell–cell communication and cell–medium nutrient exchange, which are important prerequisites for cell proliferation and signal transduction. Therefore, the 3D macrophage model steps closer to in vivo microenvironments and enables a more accurate screening of anti-inflammatory compound activities (Wang et al., 2021b). Consequently, the anti-inflammatory activities of 6,3´,4´- and 7,3´,4´-HOFL were tested on the 2D model and confirmed on the 3D model (Figure 2).

### 3.3. NO Inhibition Activity and Cytotoxicity

Anti-inflammatory efficacy and safety are partially reflected by the NO suppression activity and cytotoxicity, so they were determined on the 2D and 3D macrophages (Figure 3a–f). 6,3´,4´-HOFL was innoxious below 30 μM on 2D and below 50 μM on 3D macrophages. It showed a dose-dependent manner in NO suppression with IC_50_ values of 22.1 μM and 35.6 μM on 2D and 3D macrophages, respectively. Similarly, 7,3´,4´-HOFL was innoxious below 60 μM in 2D and 100 μM in 3D cells. The IC_50_ values were 26.7 μM and 48.6 μM, respectively.

The NO suppression activity of both flavones showed comparable IC_50_ values with the commercial anti-inflammatory agent dexamethasone, 8.34 ± 0.24 [14], and were non-cytotoxic at their IC_50_ concentrations. 7,3´,4´-HOFL was less cytotoxic but less effective at suppressing NO than 6,3´,4´-HOFL on both macrophage models, suggesting the hydroxyl group position shifting affected the potency and toxicity properties of trihydroxyflavones. Interestingly, their non-cytotoxic concentration windows were broadened by approximately 70%, but NO inhibition activities were weakened by 61% and 82% on 3D macrophages, respectively. This may be attributed to the penetration of compounds solution into the inner layer of cell clusters. The trend agrees with the observation on TC-71 that doxorubicin became less effective on 3D cells [7].

### 3.4. Enzymes and Cytokines Suppression Activities

The two flavones showed promising NO suppression activities, so their anti-inflammatory activities were further studied by examining the expressions of cytokines and enzymes in 2D and 3D macrophages.

The gene expressions of interleukin-1β (IL-1β) and interleukin-6 (IL-6) were significantly downregulated by 6,3´,4´- and 7,3´,4´-HOFL in a dose-dependent manner. At 50 or 60 μM, mRNA levels of IL-1β and IL-6 were downregulated by two flavones to the baseline on both cell models. In contrast, they failed to suppress tumor necrosis factor-α (TNF-α) at low concentrations. At higher concentrations, although 6,3´,4´- and 7,3´,4´-HOFL suppressed TNF-α transcription by approximately 80% and 66% on the 2D cells, they failed to suppress TNF-α on 3D cells. The discrepant inhibitory activities on the two cell models highlight the greater cellular resistance of 3D-cultured cells, which enables the exclusion of false positive results on 2D cells (Figure 3). Two flavones led to 48% and 35% decreases in COX-2 on 2D cells but no significant decrease in 3D cells. The iNOS inhibitory efficacy was attenuated by approximately 60% and 30% on 3D macrophages, respectively (Figure 3). The replicates of iNOS and COX-2 western blotting results are provided in Appendix A. Interestingly, 3D macrophages expressed more IL-1β, COX-2, and iNOS upon LPS stimulation, which is likely due to their greater exposure to stimulants.

### 3.5. Anti-Inflammation Mechanisms

RNA-sequencing revealed whole sets of RAW264.7 gene transcriptome altered by LPS stimulation and 6,3´,4´- or 7,3´,4´-HOFL treatment (Figure 4 and Appendix A). A total of 5764 DEGs were obtained comparing LPS with Ctrl, including 2533 upregulated and 3231 downregulated DEGs. Compared to the LPS group, 6,3´,4´- and 7,3´,4´-HOFL downregulated 461 and 411 DEGs, respectively (Appendix A).

KEGG enrichment of these DEGs showed several immune response pathways were activated by LPS and suppressed by flavones. Genes of pro-inflammatory cytokines (e.g., Il6 and Il1b), chemokines (e.g., Ccl2 and Ccl17), and receptors (e.g., Il2ra and Il13ra2) were upregulated upon LPS stimulation and downregulated when macrophages were treated with either flavone (Figure 4b). Gene transcriptions suppressed by two flavones were significantly enriched in several pathways such as “cytokine-cytokine interaction”, “influenza A”, and “rheumatoid arthritis”, as shown in Figure 4c, d, but the inflammation-related pathways are the TNF and IL-17 pathways of 6,3´,4´-HOFL and the TNF, IL-17, and JAK-STAT pathway of 7,3´,4´-HOFL.

JAK-STAT is an extensively reported inflammatory pathway, in which the dimerization of STATs is the crucial step (Figure 5) [15]. 7,3´,4´-HOFL downregulated Stat5 transcription, indicating one of its function pathways is JAK-STAT. TNF is recognized by its receptor-trimerized TNFRs, and the IL-17 cytokine family is recognized by their unique receptors IL-17RA to IL-17RD. Their recognitions induce the activation of NF-κB and MAPK pathways and further the secretion of cytokines, chemokines, and ECM remodeling factors related to inflammation (Figure 5) [16]. 6,3´,4´- and 7,3´,4´-HOFL downregulated the transcription of genes, such as Lta, Il1b, Ifnb1, and Il6, in the TNF pathway and genes, such as Ccl2, Ccl12, Mmp3, Ccl17, and Csf2, in the IL-17 pathway, as indicated in Figure 4b, suggesting their functional pathways are TNF and IL-17. The expressions of these genes were further verified by RT-qPCR (Figure 6), and the suppression potencies on these genes of two flavones were consistent with RNA transcriptome results.

### 3.6. Binding Targets of 6,3´,4´- and 7,3´,4´-HOFL

7,3´,4´-HOFL is naturally fluorescent, so this was exploited to identify its binding target. As shown in Appendix A, the c-Src band on the Western blot membrane completely overlapped with the green-fluorescent band (60 kDa) on sodium dodecyl sulfate polyacrylamide gel electrophoresis (SDS-PAGE) gel. It could be deduced that the potential receptor is c-Src, a non-receptor tyrosine kinase, due to its similar molecular weight among commonly reported flavone receptors [3].

To confirm the result, the c-Src kinase inhibitory activities of two flavones were tested. Both flavones inhibited c-Src activity in a dose-dependent manner, and their IC_50_ values were 12.0 and 20.9 μM, respectively (Figure 7c). To further understand how they bind to c-Src, the two flavones were docked with c-Src in silico. They bond into the ATP-binding pocket domain on the kinase by shape recognition and hydrogen bonds, suggesting an ATP-competitive binding manner (Figure 7). The binding amino acid residues are commonly reported in the ATP-binding pocket domain [17]. Consequently, it could be concluded that c-Src is the receptor of 6,3´,4´- and 7,3´,4´-HOFL, and its activity inhibition would lead to the downregulation of mediators in the downstream cascade.

## 4. Discussion

Oxidative stress plays an essential role in cellular redox-dependent signaling transductions to regulate cell behaviors and feedback to the host to clear ROS in normal conditions. Alternatively, sustained high levels of ROS lead to cellular damage and, finally, to some diseases [18]. Furthermore, ROS would also induce inflammation by activating the NF-κB pathway [19]. Structurally similar flavones such as chrysoeriol were reported to be antioxidants [20]. The ROS-scavenging activity of 6,3´,4´- and 7,3´,4´-HOFL was attributed to the catechol moiety (3´- and 4´ -OH) in the B ring. The flavones are able to reduce highly oxidizing radicals, such as superoxide and hydroxyl radicals, to produce a phenoxyl radical, and then, it may react with a second radical and, finally, produce a stable quinone structure [21].

IL-1β, IL-6, and TNF-α are crucial pro-inflammatory cytokines; iNOS and COX-2 are important enzymes in the inflammation cascade catalyzing the biosynthesis of NO and prostaglandins E_2_ (PGE_2_). These inflammatory mediators reflect the inflammation status of macrophages and, thus, the inflammation alleviation activity of flavone isomers. The successful dose-dependent suppression of these mediators further suggests the potential anti-inflammatory activity of 6,3´,4´- and 7,3´,4´-HOFL.

In vitro-to-in vivo translation was an inherent problem during the medication discovery. Approximately 92% of candidates failed in clinical trials, and this was mainly due to efficacy and safety problems [22]. These misleading results in in vitro examinations could be partially attributed to the divergent difference between the predominate plate-cultured and actual tissue cells. The 3D model was proven to close the gap between in vitro and in vivo examinations, minimize false positive results, and provide a reliable prediction of efficacy through better recapitulation of in vitro cell biology and growth microenvironment [13]. Therefore, the two flavones showing pro-inflammatory-mediator-suppressive activity on 2D and, especially, 3D macrophages, indicated their potential application as natural anti-inflammatory agents.

Interestingly, only the downstream genes of cytokines (e.g., Il6, Il1β, Il23, Ifnb1, and Csf2), chemokines (e.g., Ccl12 and Ccl17), and other mediators (e.g., Mmps) were suppressed, particularly those in the TNF and IL-17 pathways, rather than upstream proteins in the pathways such as NF-κB and MAPK. The phosphorylation of hallmark mediators in NF-κB and MAPK pathways was not suppressed, as revealed by the RNA-sequencing results of an insignificant difference in mapk6, mapk11, and Nfkbia genes (Figure 4b) and the Western blotting results of pp65, pp38, pJNK and pERK (Appendix A). However, both flavones downregulated the transcription of fosb, a subunit of AP-1 in the IL-17 pathway. It is advantageous that two flavones mainly inhibited downstream gene expression because, in addition to inflammatory response, upstream mediators also participate in other crucial biological activities. For example, NF-κB regulates cell survival, proliferation, and adhesion [23]. Therefore, the negligible disturbance on these mediators lowered the influence on other cellular activities and, thus, made the two flavones safer as anti-inflammatory agents.

The competitive binding to c-Src of trihydroxyflavones is partially by shape recognition, which could be attributed to the similar three-ring structure to ATP; flavones are benzopyran heterocycle linked with a phenyl, while ATP is two-ring structural adenine linked with a furan. C-Src regulates a wide array of cellular signal transduction pathways by the phosphorylation of specific tyrosine residues in other tyrosine kinases. Taking the signaling pathways into consideration in the JAK-STAT pathway, c-Src plays an important role by mediating STAT dimerization. Both JAK and c-Src are phosphorylated upon the activation of the receptor, then the activated JAK mediates the docking of STAT onto the receptor, and finally, activated c-Src leads to the phosphorylation of STAT and dimer dissociation [24]. Therefore, the bond of 7,3´,4´-HOFL to c-Src downregulated the dimerization of STAT, and finally, downregulated the expression of downstream mediators in this pathway, including CycD and c-Myc (Figure 5). In the TNF pathway, c-Src associates with the type 1 TNF receptor and JAK2 in macrophages [25]. The blockade of c-Src by 6,3´,4´- and 7,3´,4´-HOFL led to the downregulation of mediators responsible for leucocyte recruitment (Ccl2, Ccl12, and Cxcl12) and inflammatory cytokines (Il1b, Ilb, and Lta) (Figure 5).

It is worth pointing out that the flavones likely non-specifically bond to c-Src because of the simple and small chemical structure, which is supported by other less enriched green-fluorescent bands on the SDS-PAGE gel (Appendix A). Furthermore, the current 2D and 3D in vitro studies were conducted on murine macrophages. However, human cell lines are closer to replicating human physiological conditions, so in the future, 3D human macrophage models such as microglia should be established to better bridge the 2D in vitro and clinical studies and, thus, provide more predictive results of antioxidant and anti-inflammatory activities.

## 5. Conclusions

In summary, 6,3´,4´- and 7,3´,4´-HOFL exhibited significant anti-inflammatory and cellular antioxidant activities. They suppressed the overexpression of pro-inflammatory biomarkers on both 2D and 3D macrophage models. They both exert anti-inflammatory activities through the IL-17 and TNF signaling pathways and 7,3´,4´-HOFL also through the JAK-STAT pathway by binding to c-Src in RAW264.7. Our results indicate that natural flavones may be developed, as nutritional supplements with clear mechanisms, to alleviate inflammatory diseases. Particularly, the natural fluorescent property of 7,3´,4´-HOFL could be utilized to study the cellular receptors and metabolisms of flavones.

## Figures and Tables

**Figure 1 antioxidants-12-00204-f001:**
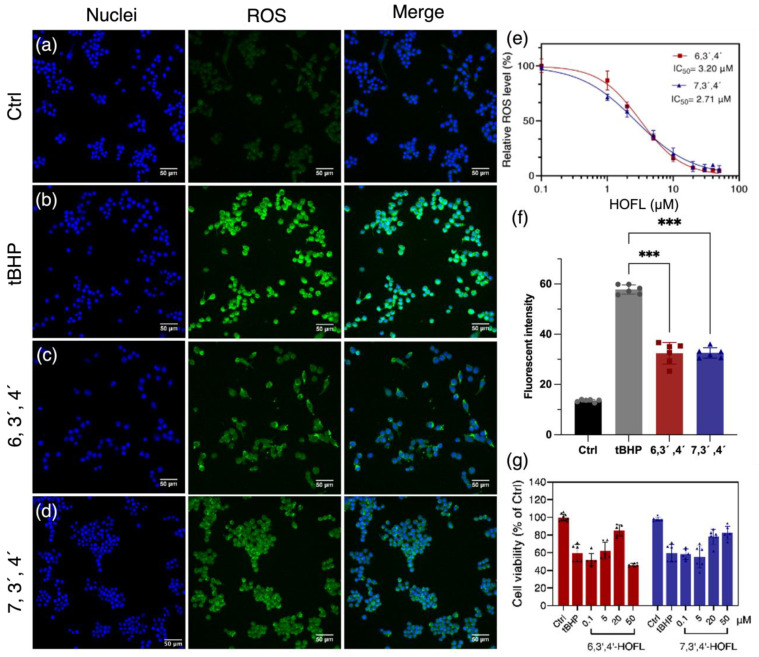
ROS-scavenging activity of 6,3´,4´- and 7,3´,4´-HOFL in tBHP-induced RAW264.7. Fluorescent microscopy images of untreated (**a**), tBHP-induced (**b**), and tBHP- and 6,3´,4´-HOFL- (**c**) or tBHP and 7,3´,4´-HOFL-treated cells (**d**). Blue: cell nuclei stained with H33342; green: ROS stained with H2DCFDA probe. Dose–response curves of cellular ROS treated with tBHP and flavones (**e**); fluorescence intensity of ROS of microscopy images (**f**), *** *p* < 0.001 vs. tBHP; and cell viability induced by tBHP and treated with flavones (**g**).

**Figure 2 antioxidants-12-00204-f002:**
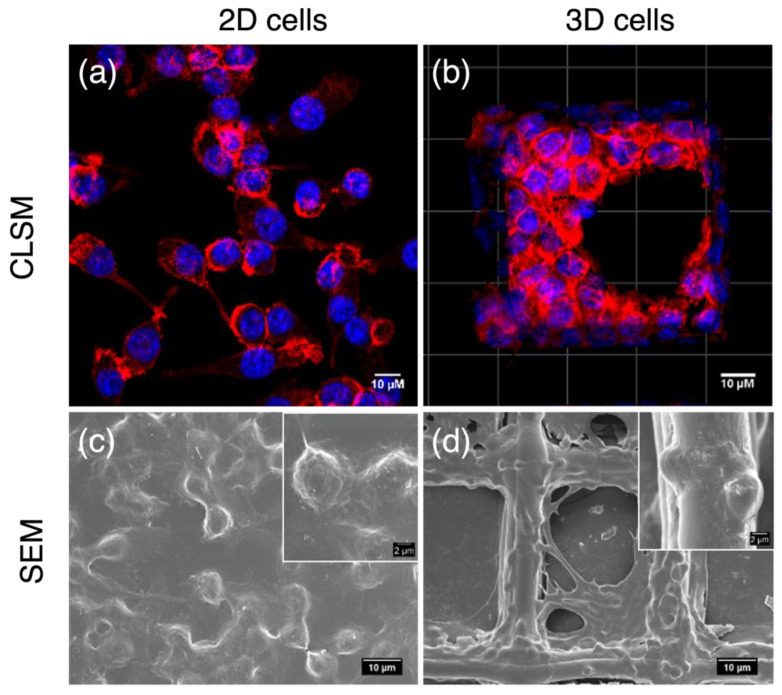
Morphologies of 2D- and 3D-cultured RAW264.7 cells. CLSM and FESEM images of 2D macrophages (**a**,**c**); CLSM and SEM images of macrophages cultured on PCL scaffolds (**b**,**d**). Blue: cell nuclei stained with H33342; red: F-actin stained with F-actin cytopainter.

**Figure 3 antioxidants-12-00204-f003:**
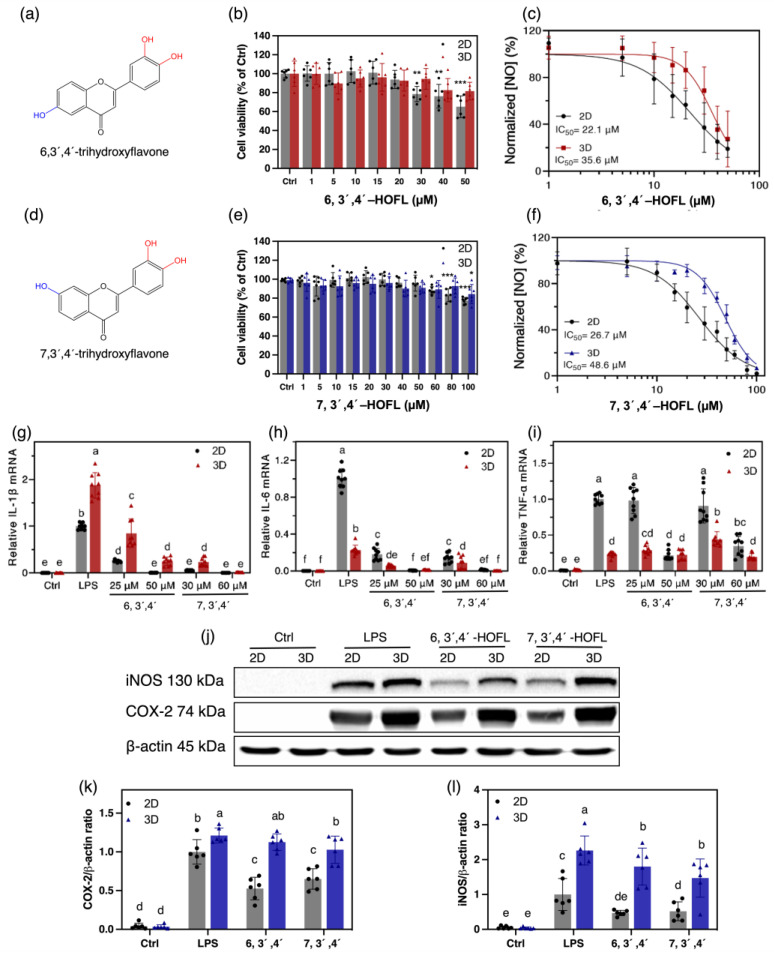
Cytotoxicity and anti-inflammatory activities of 6,3´,4´- and 7,3´,4´-HOFL on 100 ng/mL LPS-induced 2D and 3D RAW264.7 cells. Chemical structures (**a**,**d**); cytotoxicity (**b**,**e**); and dose–response curves of NO suppression (**c**,**f**); n = 6. The inhibitory activity on the gene expressions of IL-1β (**g**), IL-6 (**h**), and TNF-α (**i**) on 2D and 3D cells; n = 9. Inhibitory activities of 6,3´,4´- (50 μM) and 7,3´,4´-HOFL (60 μM) on the protein expressions of iNOS (**k**), COX-2 (**l**), and their bands (**j**); n = 6. The data are shown as mean ± SD. * *p* < 0.05, ** *p* < 0.01, *** *p* < 0.001, and lowercase letters represent the differences between the sample groups (*p* < 0.05).

**Figure 4 antioxidants-12-00204-f004:**
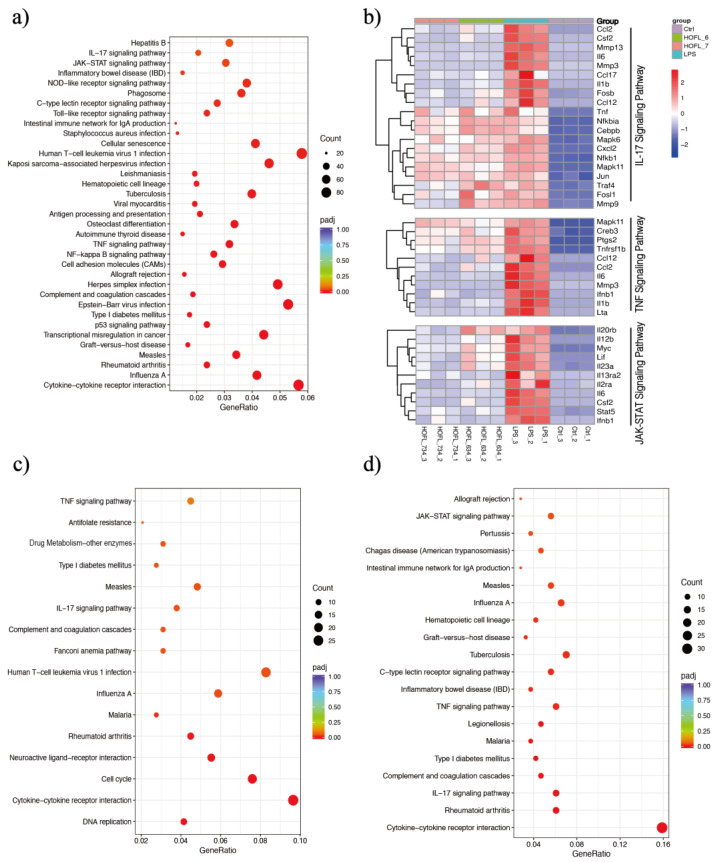
6,3´,4´- and 7,3´,4´-HOFL exert anti-inflammatory activity via IL-17, TNF, and JAK-STAT pathways. KEGG enrichment scatter plots of DEGs of LPS vs. Ctrl (**a**), 6,3´,4´-HOFL vs. LPS (**c**), and 7,3´,4´-HOFL vs. LPS (**d**). Clustering heatmap of DEGs involved in IL-17, TNF, and JAK-STAT pathways (**b**).

**Figure 5 antioxidants-12-00204-f005:**
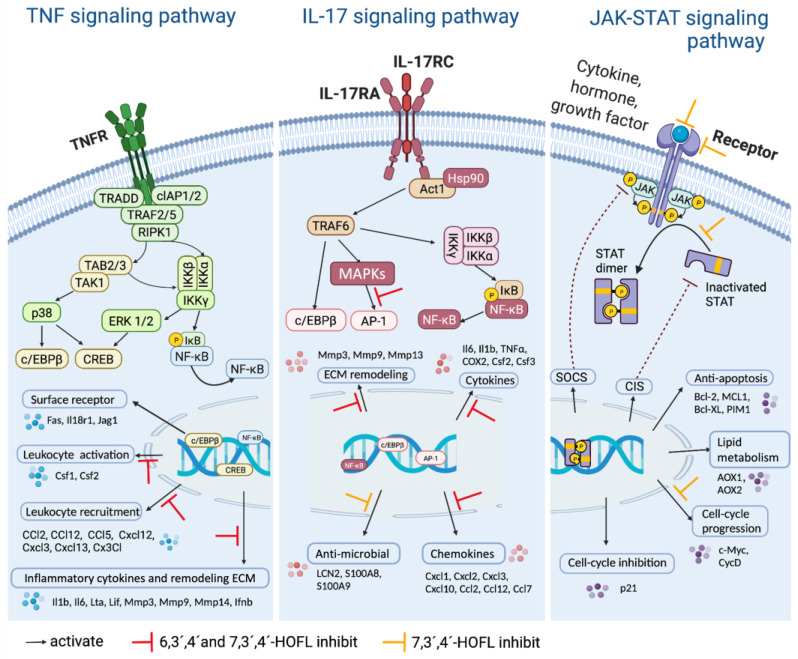
Schematic diagram of the signaling pathways. Created with BioRender.com.

**Figure 6 antioxidants-12-00204-f006:**
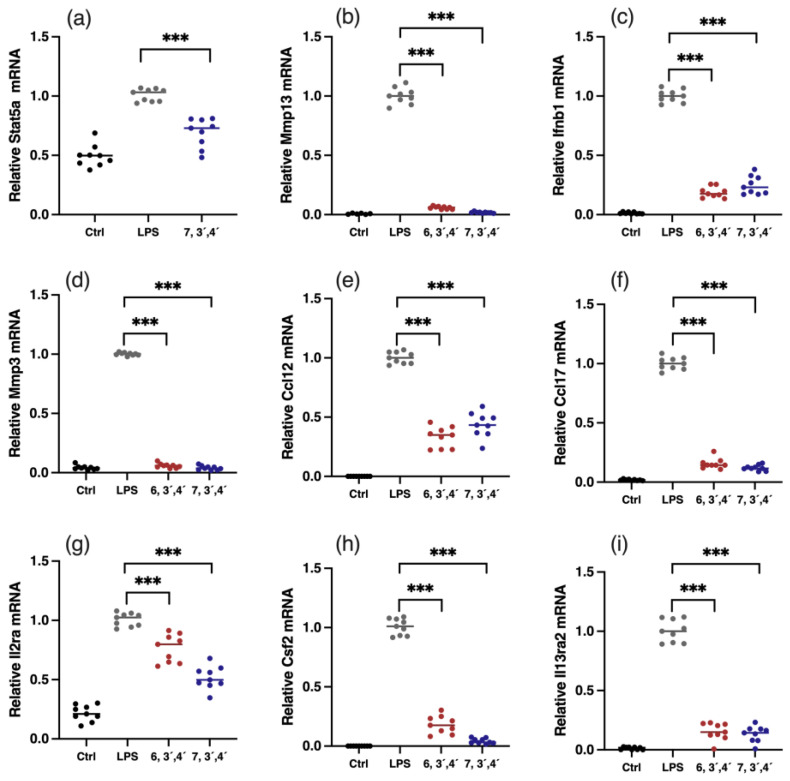
Verification of gene expressions of key DEGs. Relative mRNA expression of Stat5 (**a**), Mmp13 (**b**), Ifnb1 (**c**), Mmp3 (**d**), Ccl12 (**e**), Ccl17 (**f**), Il2ra (**g**), Csf2 (**h**), and Il13ra2 (**i**). Data are shown as mean ± SD, n = 9, *** *p*< 0.001.

**Figure 7 antioxidants-12-00204-f007:**
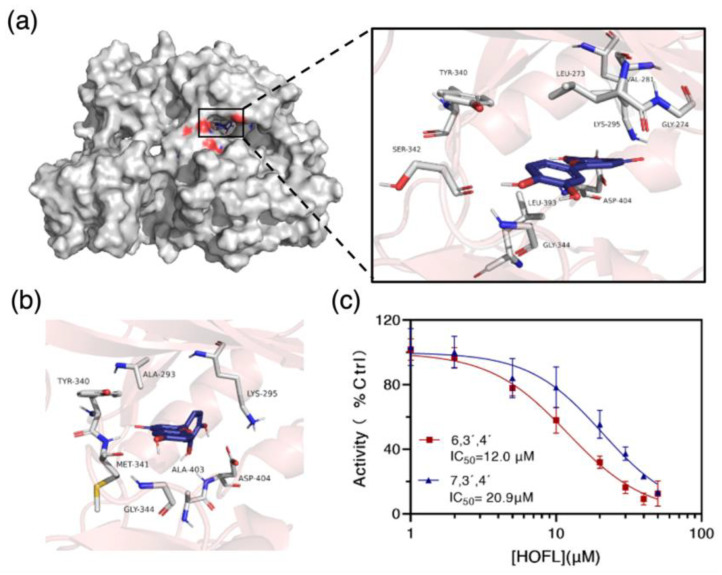
The binding target of 6,3´,4´- and 7,3´,4´-HOFL. Computational modeling of 6,3´,4´-HOFL bonded to the ATP-binding pocket on the c-Src (**a**); the detailed binding scenario of 7,3´,4´-HOFL (**b**); and dose–response curves of 6,3´,4´- and 7,3´,4´-HOFL on c-Src activity (**c**). Data are shown as mean ± SD, n = 3.

## Data Availability

Data is contained within the article, Appendix A.

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
