# Peer review of "Antioxidant and Anti-Inflammatory Effects of 6,3’,4´- and 7,3´,4´-Trihydroxyflavone on 2D and 3D RAW264.7 Models"

_antioxidants, 2023, doi:10.3390/antiox12010204_

Round 1
Reviewer 1 Report
The interesting paper by Huang and coworkers describes the antioxidative and anti-inflammatory effects of two flavones using in vitro 2D and 3D macrophage models. The experimental procedures have been carried out in a reliable way and the interpretations offered are all very reasonable. There are a few minor points as indicated below that need to be addressed by the authors before publication in the journal.
1. Line 16: Is “by via” correct?
2. Line 331: “The ROS activity” should be “The ROS-scavenging activity.”
3. Line 333: “a phenoxyl radical” is better rather than “an aroxyl radical.”
4. Line 348: Are the two flavones novel?
Author Response
Thank you so much for pointing out the errors. They have been corrected as indicated below.
- Line 16: The redundant "via" was deleted.
- Line 338 (previous 331): “The ROS activity” has been corrected to “The ROS-scavenging activity.”
- Line 340 (previous 333): “an aroxyl radical” has been replaced by “a phenoxyl radical”.
- Line 355 (previous 348): The "novel" has been deleted.
Reviewer 2 Report
One of the better manuscripts I have had the enjoyment and pleasure of reading recently. It is a comprehensive study, utilising the appropriate methodologies as well as presenting the important paradigm of 3-D organotypic modelling. The paper is well written in a clear and concise manner. The hypothesis is clear and well defined, being supported with topical and relevant citations. The data are well interrogated, interpreted, and presented. The conclusions drawn are logical and important. There findings add to our critical knowledge of this area, and I foresee the paper having a wide readership, as well as having the potential to be well cited. I commend the authors on an excellent study.
Minor points only-
1. Please define abbreviations (e.g. tBHT, PCL scaffolds, EHDJ printer)
2. For more clinically relevant end-points, future studies would benefit with a human macrophage cell line. A brief narrative on this would be good in the discussion.
3. Some minor syntax/grammatical errors e.g. Ln 34-35; However, sustained high level of ROS leads to inflammation via releasing inflammatory mediators and the macrophage activation. Ln52-53; 7,3 ́,4 ́-HOFL showed preliminary anti-inflammatory activity only by suppressing NO, but their signaling pathway is not clearly revealed yet. Ln59; genes expression.
4. Ln154 to Ln156; equation missing place correct. "T1=1500 s, T2=1900 s 155 R: reaction rate; TF: test flavones; BC: background; NI: no inhibitor." Ln 34-35
Author Response
Thank you so much for the valuable suggestions. We have revised the manuscript accordingly.
- All the abbreviations have been defined when they first appear and also been listed at the end of the main text.
- The discussion on human macrophage has been added in line 394-399.
- The syntax/grammatical errors have been corrected.
- The missing equations have been added.